# Elite-CAM: An Elite Channel Allocation and Mapping for Policy Engine over Cognitive Radio Technology in 5G

**DOI:** 10.3390/s22135011

**Published:** 2022-07-02

**Authors:** C. Rajesh Babu, Amutha Balakrishnan, Kadiyala Ramana, Saurabh Singh, In-Ho Ra

**Affiliations:** 1Department of CSE, SRM Institute of Science and Technology, Chennai 603203, India; rajeshbc@srmist.edu.in (C.R.B.); amuthab@srmist.edu.in (A.B.); 2Department of IT, Chaitanya Bharathi Institute Technology, Hyderabad 500075, India; ramana.it01@gmail.com; 3Department of Industrial and System Engineering, Dongguk University, Seoul 04620, Korea; saurabh89@dongguk.edu; 4School of Computer, Information and Communication Engineering, Kunsan National University, Gunsan 54150, Korea

**Keywords:** cognitive radio, wireless communications, 5G CORE, spectrum allocation, 5G communications

## Abstract

The spectrum allocation in any auctioned wireless service primarily depends upon the necessity and the usage of licensed primary users (PUs) of a certain band of frequencies. These frequencies are utilized by the PUs as per their needs and requirements. When the allocated spectrum is not being utilized in the full efficient manner, the unused spectrum is treated by the PUs as white space without believing much in the concept of spectrum scarcity. There are techniques invented and incorporated by many researchers, such as cognitive radio technology, which involves software-defined radio with reconfigurable antennas tuned to particular frequencies at different times. Cognitive radio (CR) technology realizes the logic of the utility factor of the PUs and the requirements of the secondary users (SU) who are in queue to utilize the unused spectrum, which is the white space. The CR technology is enriched with different frequency allocation engines and with different strategies in different parts of the world, complying with the regulatory standards of the FCC and ITU. Based on the frequency allocation made globally, the existing CR technology understands the nuances of static and dynamic spectrum allocation and also embraces the intelligence in time allocation by scheduling the SUs whenever the PUs are not using the spectrum, and when the PUs pitch in the SUs have to leave the band without time. This paper identifies a few of the research gaps existing in the earlier literature. The behavioral aspects of the PUs and SUs have been analyzed for a period of 90 days with some specific spectrum ranges of usage in India. The communal habits of utilizing the spectrum, not utilizing the spectrum as white space, different time zones, the requisites of the SUs, the necessity of the applications, and the improvement of the utility factor of the entire spectrum have been considered along with static and dynamic spectrum usage, the development of the spectrum policy engine aligned with cooperative and opportunistic spectrum sensing, and access techniques indulging in artificial intelligence (AI). This will lead to fine-tuning the PU and SU channel mapping without being hindered by predefined policies. We identify the cognitive radio transmitter and receiver parameters, and resort to the same in a proposed channel adaption algorithm. We also analyze the white spaces offered by spectrum ranges of VHF, GSM-900, and GSM-1800 by a real-time survey with a spectrum analyzer. The identified parameters and white spaces are mapped with the help of a swotting algorithm. A sample policy has been stated for ISM band 2.4 GHz where such policies can be excited in a policy server. The policy engine is suggested to be configured over the 5G CORE spectrum management function.

## 1. Introduction

The 5G wireless structure desires to support varied service models, different delays, and reliable demands. The 5G broadband wireless communication results in several issues relevant to real-time radio resource management. For example, ultra-reliable transmission demands real-time applications of wireless infrastructure for delay and reliability. The model of an autonomous wireless network with simultaneous service delivered in actual timing can be attained by enhanced changes to the recent deterministic control and optimization of radio resource management models.

Cognitive radio resource management necessitates a close linkage between spectrum management capability and software-defined radio properties, i.e., physical layer-supported modes of operation. The concept behind distributed artificial intelligence (DAI) is to shift from individual to collective behavior in order to overcome the limitations of traditional AI when solving complex problems that require the distribution of intelligence across multiple entities. DAI conducts research in three key areas: distributed problem solving, parallel artificial intelligence, and multi-agent systems. AI and 5G mobile technology have enormous potential to greatly improve profitability, productivity, and efficiency in many areas of business and society, allowing the creation of previously imagined products and services. The combination of AI with 5G will have a significant impact on growing industries like agriculture, healthcare, and education, despite the fact that mainstream applications have not yet materialized. Interest in 5G networks is growing quickly although many mobile operators are still recouping their investments in older network standards. 5G networks are able to transport more data every second and per hertz of the spectrum than previous generations of cellular networks because of improved spectral efficiency. Due to the limited supply and expensive cost of spectrum, this is critical. More concurrent users may be served at a lower cost in order to improve spectral efficiency. Automation and machine-enabled decision-making will transform almost every area of everyday life as a result of breakthroughs in emerging technologies, the rise of artificial intelligence, and higher data speeds. AI and 5G are undeniably the driving forces behind this next technology revolution. However, the vast majority of AI-based 5G use cases are not really 5G use cases since they do not need a technological transition. Apps with a response time of less than one millisecond will stand out because of their decreased latency. In emerging economies, the AI and 5G journey will very certainly require strengthening existing use cases and developing new ones that are not served by present technology. Combining artificial intelligence (AI) with 5G’s higher data speeds opens up a slew of new possibilities. The following are among them: Enhancements to mobile communications networks and services. Mobile communication service providers face increasing challenges. In order to deal with the complexity of 5G networks and the billions of Internet of Things (IoT) devices they can support, the deployment of 5G networks is significantly more difficult than in previous generations due to the upgrades required for radio, edge, transport, core, and cloud infrastructure. All of these upgrades should be managed optimally by AI. 5G deployment is being hampered by a lack of available spectrum. The frequency at which 5G operates determines its power and speed. Utilizing frequencies with smaller bandwidths that have advantageous characteristics is one way to get around spectrum limitations. When it comes to 5G, a frequency of 600 MHz may not deteriorate quickly and may still be capable of reaching devices even if they are encased in columns [1].

Here, dynamic spectrum access used by CR tools demands abandoning present pre-programmed strategies. A policy-based model for CR platforms and a new policy reasoning structure for the same have been applied for envisaging policies for the CR system. The essence of CR [2], reconfigures the use of policies from radio devices, and as a result, the adaptive as well as flexible nature of radio systems yields optimized use of spectrum resources. Hence, the strategies are exploited and dynamically modified by shareholders.

The development of a policy-based spectrum allocation system for 5G networks is crucial, thus we built:An elite channel selection system that determines the parameters that must be selected for successful spectrum allocation from an exhaustive collection of parameters.A new parameter mapping technique for translating the needed channel specification to available spectrum holes.A POMDP-enabled policy management system that provides secondary users with service based on pre-mapped channels.

The remaining sections of the paper are organized as follows. Work on 5G networks, channel selection and allocation systems, and policy-based systems are all discussed in Section 2. The policy engine framework is presented in Section 3. We demonstrate the parameter mapping and POMDP-enabled reward methods in Section 4 and Section 5, and the findings and testing performance in Section 6. In Section 7, we will also share our findings.

## 2. Related Work

An exhaustive survey on cognitive radio sensing and allocation has been carried out and the summary of the survey is given in Table 1.

### 2.1. Survey on Spectrum Sensing

Ahmad, W.S et al. [3] studied the most current 5G enablers and SS (recent spectrum sharing) technologies in detail. In order to better understand the importance of 5G networks, SS methods were categorized and SS surveys and related research on SS techniques were analyzed. One of the main SS techniques is based on network design, spectrum allocation behavior, and the manner of spectrum access. Cognitive radio technology with SS was also thoroughly investigated in relation to 5G rollout. There are discussions on the existing deployment of SS and CR and the steps to allow efficient 5G progress for a complete examination. Dinh-Thuan Do et al. [4] showed off a unique system design that combined the benefits of both a NOMA and CR network with the Simultaneous Wireless Information and Power Transfer technology. Downlink spectrum sensing and wireless energy harvesting are the primary functions of a radio spectrum unit (RSU). Remote vehicle outage probabilities may be accurately expressed using CR-enabled NOMA systems. Overlay CR-enabled NOMA network performance is severely limited by the transmit power and the number of RSU units designed. Simultaneous gains from energy collection and RSU selection were enabled by the CR, according to the findings. KedirMamo Besher et al. [5] described the Sensing Primary User Interface (SenPUI), a brand new cognitive radio algorithm. During the dynamic inactive phase of communication, an energy scan was conducted. It is therefore suggested that an application packet-based main user ID be used as the foundation for Primary User Interface (PUI) prevention. As a result of this paper’s recommendations, the target issues have been decreased by an average of 10–30%. The memory, battery life, and size limits of wireless sensor networks (WSNs) were taken into account while developing and implementing new solutions. Vicent Pla et al. [6] developed a set of Markovian models that may be used to analyze and evaluate spectrum sensing tactics in various scenarios. Markov phase renewal is used to describe the alternating idle and busy times of a channel’s occupancy by main users. Secondary users’ activity is shown by the length of transmissions, sensing periods, and idle intervals between subsequent sensing periods. These durations are modeled using phase-type distributions, allowing the model to be very flexible. Models for both finite and infinite queues are developed using the Markovian arrival process, which depicts the arrival of secondary users. Additionally, provided was a basic explanation of how an SU transmission restarts after being stopped by a PU action in the suggested models. Many essential performance measures in cognitive radio networks are established by a thorough study of the suggested models. Despite their generality and flexibility, the proposed models may be numerically evaluated.

### 2.2. Survey on Spectrum Allocation and Algorithms

Yung-Fa Huang et al [7] used the hidden Markov model (HMM) to examine cooperative spectrum allocation in cognitive radio (CR) networks for opportunistic spectrum access (OSA). Slot-by-slot spectrum sensing was performed, assuming that the main channel operates in TDMA mode. Although typical Bayesian updating relies on a single observation, our method concatenates all of the data from secondary users (SUs). In the suggested technique, the channel activity was determined using a predetermined threshold based on belief. In contrast to the simple majority voting method, which did not have a threshold, the suggested approach was more flexible in system functioning. To test the proposed concatenated update approach, a simulation was run using the number of SUs and the majority vote strategy. The simulation indicated that a busy state and an idle state can be reliably detected at around one per SU. METIN OZTURK et al. [8] allowed CRNs to be made aware of unlicensed user needs while also reducing sensing latency. As a consequence, a unique QoS-based optimization phase and two distinct decision methods were presented. Artificial neural networks (ANNs) were used to predict future radio access technology (RAT) traffic loads in various frequency bands (ANNs). Assumptions based on this information have led to two possible solutions. The original approach was only concerned with reducing latency. As a result, it was conceivable to create a virtual wideband (WB) sensing solution by using traffic loads in WB to enable narrowband (NB) sensing. The second strategy, which was based on Q-learning, tried to reduce sensing latency while still meeting other user needs. Random selection’s sensing latency was lowered by 59% with the first technique whereas Q-learning helped boost satisfaction by 95.7% with the second method.

### 2.3. Survey on Learning-Based Allocation

Magorzata Wasilewska et al. [9] recommended the use of k-nearest neighbors and random forest as two machine learning (ML) approaches for improving spectrum sensing performance. By using methods such as energy detection (ED) and data based on energy vectors (EVs), a new 5G radio system was able to make use of the resource blocks made accessible by the previous 4G LTE network. Time, frequency, and location relationships were exploited by the algorithms. If the input training data set is correctly selected, ML algorithms may considerably increase spectrum sensing performance. The benefits and drawbacks of using input data sets with ED assessments and energy values in the actual world were investigated. According to ZHAOYUAN SHI et al. [10], the massive multiple-input and multiple-output that underlays cognitive radio user selection are always QoS-aware. There are two major ways in which a CR may be implemented: the CSI of any cross-network is inaccessible at the secondary base station, but the secondary base station (SBS) has access to the CSI of the cross-channel channel state. Low-complexity algorithms for increasing users while using the least amount of power (IUMP) and methods for decreasing users while using the most amount of power (DUMP) were developed to solve user selection with power allocation. The intractable challenge was addressed using a deep reinforcement learning-based method, enabling the SBS to accomplish effective and intelligent user selection. These algorithms outperform current user selection approaches dramatically in simulations. The neural network was able to rapidly learn the best user selection strategy in an unknown dynamic environment with a high success rate and fast convergence, as the findings also revealed. Arunthavanathan et al. [11] demonstrated that a Markov decision process (MDP) might be used to make decisions based on the current transmissions in the channel. Decisions were made and the effects of interference and waste were evaluated for a range of occupancy rates. Comparative studies of the partially observable Markov decision process (POMDP) and the Markov decision process (MDP) were also conducted. Reinforcement learning and policy-based multi-agent systems were suggested by Jaishanthi et al. [12]. Because the system is powered by AI, the nodes are able to make autonomous choices about which channel to use and how to move between multiple channels because they have all of the necessary information stored in the repository. Maximizing the simulation’s use of the spectrum yielded better outcomes.

## 3. Policy Management for Cognitive Radio

In this section, we explore and understand the fundamentals of policies to be defined for channel allocation and channel mapping. Policies are declarative statements that describe the administrative norms that different organizational bodies follow. There is a business basis for automatic policy management of resources in network and information system administration. Policy management allows you to define organizational goals that can be read and enforced by the network as part of the policy management process [13]. In order to meet administrative objectives and constraints on privacy, allocation of resources, app prioritization, and service quality via the use of automated management rules, it is possible to alter network device settings at runtime. Due to the existing policy’s centralized, static character, wireless communication is confronted with two important issues: spectrum shortages and implementation delays. The following capabilities are required to achieve opportunistic spectrum access:Sensing across a broad frequency spectrum and primary channel connection establishmentDefining potential opportunitiesCoordinating the usage of recognized opportunities via communication across devicesDefining and enforcing interference-restricting regulations

Enforcing compliance with relevant regulations while capitalizing on highlighted possibilities. Devices may be made to load and alter their policies at any time by using declarative language that is based on formal logic. Instead of relying on hardware, firmware, or device-specific software to implement rules, our solution allows devices to be flexible and respond to policy changes. Decoupling policy formulation from implementation and optimization decisions specific to individual devices is the basis for this policy-based approach to radio operation. For starters, there’s less need for certification work. In order to be accredited, a device must have the ability to dynamically load the rules that control its behavior. It is possible to independently validate the policy reasoner and every rule and then test the configuration of devices to ensure that they appropriately understand the outputs of the policy reasoner. In order for a policy reasoner to be effective, it must have a policy language that is sufficiently expressive and methodologies for analyzing and automating policy enforcement.

### 3.1. Policy Engine

To govern network resource behavior, a computer or process must be capable of consuming and applying machine-readable rules. The computer/process is referred to as the policy engine, as in Figure 1. For events requiring changes to the configuration of a resource, the policy engine is responsible for providing a response. Configuration directives or authorizations tailored to specific network devices are often the results of the policy engine. The policy engine as shown in Figure 1 connects domain-specific goals with the capabilities of the devices they are associated with [14]. In spite of the fact that policy engines must interact with particular vendor devices, they have been seen as general-purpose instruments susceptible to logical thinking through rules. Strategic reasoner (SR) and rule-based reasoner (RBR), the policy engines for cognitive radio, work together to determine if spectrum access options are available that are compatible with the currently active set of regulatory requirements. The SR sends a channel request to the RBR with a transmission plan it has developed based on the behavior data it has collected from the channel’s primary users and its present state [15,16]. For the purpose of making sure the allocation plan is in compliance with the rules, the RBR compares channel requests to those policies. The RBR provides a denial-of-service response to the SR if it detects that a portion of configuration states in the channel request is erroneous. Constraints for opportunistic access are the adjusted parameters derived by the RBR. The basic reasoning challenge in policy-enabled cognitive radio is to determine whether a channel request should be processed or refused based on a given set of rules and allocation techniques, and, if denied, to calculate the limitations for opportunistic access to the channel.

### 3.2. Assumptions for Elite Channel Allocation Algorithm

It is impossible for hardcode regulations allowing secondary users to use available spectrum holes across several radio bands that fluctuate in frequency and location.The spectrum occupancy denoted by Γ by primary users, is given by [17,18]:



(1)
$$\Gamma =\left(1/\mathrm{KN}\right)\sum_{K=1}^{K}\sum_{N=1}^{N}\Gamma f\left(n\right),t\left(k\right)$$



where total number of operating frequencies in a band is denoted by N and K denotes the number of total samples associated with each point of frequency fb [(99 kHz–3 GHz), (100 MHz–1000 MHz) (50 MHz–4400 MHz)]The anticipated spectrum characteristic needs for secondary users (bandwidth Ba, power dB req, time access period, SNR) for various types of spectral intensity, geographic location, and regulatory authority [11]


(2)
$$R\left(x\right)=\sum_{n=0}^{n-1}\sum_{m=0}^{m-1}\left({\mathrm{ch}}_{\mathrm{mn}}B_{\mathrm{nm}}\log_{2}\frac{{\mathrm{Pow}}_{\mathrm{mn}}}{\lambda }\cdot \frac{{\mathrm{PoG}}_{\mathrm{mn}}}{\mu }\right)$$


where R(x) denotes SU utility function,ch_mn_ channel m utilized by node nB_mn_ bandwidth of channel m utilized by node nPow_mn_ power required in dB for channel m utilized by node nPoG_mn_ power gain in dB for all fading and path loss over the channelλ and µ denotes SNR and minimal threshold power required to transmit in the channel.The bandwidth available is divided into a predetermined number of channel bands.Each channel is optimal.At any one time, each channel must serve a single user (primary user or secondary user).Secondary user time sensing and synchronization are intended to occur at guard band frequency.At guard band frequency, time sensing and hand off to the primary user from the secondary user are anticipated.Each user’s entrance is a distinct event that occurs independently of subsequent users. Distribution of Poisson.The number of available channels is more than the number of PUs.Primary users are authorized users who take precedence over secondary users.

Figure 2 and Table 2 summarize the set of exhaustive cognitive radio parameters for transmitters, receivers, and channels. The parameters [19,20] are considered by the swotting algorithm which is mentioned in Algorithm 1, for doing parameter mapping before channels are mapped.


**Algorithm 1.** Elite-CAM—Algorithm for Cognitive Radio Channel Selection and Allocationfor All Channels,
   SetIdle       {
If channel Ch is IDLE at time T_a_
   If authorized
    Check Primary User back-off time B_ta_
    If available IDLE for Time T_b-a_, Mark Channel specification Γ, and release for Secondary User    Channel Specification as in Equation (1)         $\Gamma =\left(1/\mathrm{KN}\right)\sum_{K=1}^{K}\sum_{N=1}^{N}\Gamma f\left(n\right),t\left(k\right)$

      N—Number of frequency points in the band 
      K—Number of time samples for each frequency point
} Repeat SetIdle for all channels;

For all Idle Channels
{
AllocateSU
Check SU Queue
   If unauthorized
     Transfer Channel specifications      $\Gamma_{1}$, $\Gamma_{2}$ … $\Gamma_{N}$ and
     Idle Time Slots—T_b−a_($\Gamma_{1}$), T_b−a_($\Gamma_{2}$), … T_b−a_($\Gamma_{N}$), 
      to SU 
   Identify the QoS requests from Secondary User
For
       $R\left(\mathrm{SU}1\right)=\sum_{n=0}^{n-1}\sum_{m=0}^{m-1}\left({\mathrm{ch}}_{\mathrm{mn}}B_{\mathrm{nm}}\log_{2}\frac{{\mathrm{Pow}}_{\mathrm{mn}}}{\lambda }\cdot \frac{{\mathrm{PoG}}_{\mathrm{mn}}}{\mu }\right)$
If $\Gamma_{x}$ satisfies the requirements R(SUy)

Allocate
Ch_x_ to SUy with B_x_ Bandwidth of channel m utilized by node y, Pow_x,_ the Power required in dB for channel m utilized by node y, PoG_x_ Power Gain in dB for all fading and path loss over the channel

} Repeat AllocateSU for all Idle Channels
When PU arrives,
   If the channel is occupied by SUz,
    Pre-empt SUz from Channel Chz and release for PU
For SUz, call AllocateSU and identify spectrum hole for re-allocation
end


## 4. Elite Mapping—Swotting Algorithm for Parameter Mapping Repository

The swotting algorithm given in Algorithm 2, is a reinforcement Q-learning algorithm that determines decision-making policies without comprehensive modeling of the radio environment. This suggests that Q-learning describes and improves the performance measures of interest, rather than addressing network performance-related aspects such as wireless channel state and mobility, for instance. It has four parameters: states, action a, probabilistic transition function Qr, r, and reward function Sb, b. The state may represent internal events occurring inside the agent, such as the size of the instantaneous queue, or external phenomena occurring outside the agent, such as the agent’s use of the wireless medium. The reward function shows the system’s response to the quality of its actions, and as a result, the system acquires experience. At time t, the agent observes the environment’s state st. The action cat is chosen in accordance with the state cst. According to cat and Qr, r, the environment changes the state and reward function rt = R(cst, cat) obtained as a consequence of the change is recorded and supplied back to the agent. The optimum Q-value is a measure defined for each state-action pair in the process of determining the best policy s, and it is assessed in the following manner [21,22].
(3)$$\mathrm{SWQ}(a,s)=E\left\{P\left(a,s\right)\right\}+\gamma \sum_{a\in s}^{}P_{a,a},max_{c\in A}\mathrm{SWQ}\left(a^{\prime },c\right)$$
where a, s—available state sets,

γ—Factor for discount

The ideal policy can be identified using the following criteria [12]:(4)πs=arg max a∈A SWQ(a,s)

The elite swotting algorithm finds SWQ (a,s) in a recursive manner using the following rule [19]
(5)$$\mathrm{SWQ}\left(a,s\right)=\left(1-\alpha \right)\mathrm{SWQ}\left(a,s\right)+\alpha (\gamma \left(a,s\right)+{\gamma \max}_{b\in A}\mathrm{SWQ}\left(a^{\prime },b\right))$$

Appropriate actions are rewarded with a rise in their Q-value. In comparison, an incorrect action results in a penalty and a fall in the Q-value. The Q-value is stored in a size-dependent two-dimensional lookup Q-table. It was shown that when each state-action pair is visited infinitely often, this updated algorithm converges to the optimum Q-value. Thus, learning is composed of two processes that govern the selection of actions: exploitation and exploration [23]. Exploitation is the process of choosing the optimum action based on previously discovered optimal policies, whereas exploration is the process of randomly picking nonoptimal actions and finding new ones. The exploration rate E determines the balance between exploration and exploitation.
**Algorithm 2.** Elite Mapping Swotting Algorithm for Parameter Mapping**Prerequisites:** present state cs(t), previous state ps(x), and F(t)
Ascertain that the Swotting algorithm selected produces the maximum possible output. OP 
**Training:** given the state of the network cs(t);
Probabilistic exploration γ;
Choose one action at random;
New Update UP(t)={b|W(cs(t),b)=1} for cs(t);
Probabilistic Exploitation is 1 −γ;
Choose α records UP”(CS’(ps(x),F(t)) from actions F in accordance with CS’(ps(x),F(t))
Resolve Re(cs(t), r) in accordance with next r(x) and fill UP^R^(cs(t));
**if**
$(y*\;\mathrm{exist}=\max_{y}(y|\mathrm{cs}(y)\;\in$ UP”(CS’(cs(t)),F(t)) ∩ UP’(cs(t)) then
Choose the action r(x);
**Else if**
Choose action from UP^R^(cs(t)); 
**end if**

do update UP(cs(t), r) 
do
repeat SWQ;
while (t = t + 1 and s = s.t + 1)

### Parameter Mapping Repository (PMR)

Assisting the cognitive network, the PMR accepts sensory input from the communication layer, selects suitable channels [24,25], and fine-tunes transmission parameters to enhance the cognitive network’s performance for each generic interface; the PMR is responsible for storing and processing sensory data that has been sent to it as in Figure 3. In addition, it prepares and sends various transmission parameter setting options through the corresponding interfaces along the stack levels. The parameter mapper is responsible for making the final choice of temporary data storage repository for parameter mapper modules’ data flow. Following are the modules that make up the parameter mapper framework.

Negotiator Module: It serves as a coordinator for the decision-making layer’s duties. The decision module receives sensory data from the repository and uses it to make a choice. Negotiators may also use policy queries to check whether the transmission parameters they’ve selected are in compliance with any constraints as in Table 3, placed by the policy layer. Following receipt of the policy-layer answer, the negotiator communicates the choice to the repository’s action module.

Resolver Module: It accepts requests from the negotiator and performs the following activities. Determine whether there is a greater priority in the case that many channel requests are pending. After the spectrum sensing data from the spectrum sniffer is gathered, the process of allocating channels starts. As a final step, a decision module runs the optimization process and passes the policy-validated transmission parameters to the action broker.

## 5. PMDP Model for Channel State and Reward

In a Markov decision process (MDP), [26] decisions are made based on the most current state data in a discrete-time stochastic management process as in Figure 4. Current state is S(t) ∈ S, where S implies the complete state space. In real-time application, the recent states are adaptable to conditions, and partially observable MDP (POMDP) has been applied for computing the decision policy according to the partially accessible data as well as observations from the physical atmosphere. Generally, optimization models are applied for gaining the solution for POMDP-related issues.

Consider that $c_{r}^{1}\left(i\right)$(t) implies the condition of rth channel of ith carrier in time slot t, the status of complete system in tth time slot can be expressed as [13]
(6)S(t)={C(1)(t),…,C(i)(t),…,C(N)(t)},∀S(t)∈S and
(7)C(i)(t)={c(i)1(t),…,c(i)r(t),…,c(i)R(i)(t)},∀c(i)r(t)∈{0,1}
where *C*(*i*)(*t*) defines the channel set of *i*th carrier from time slot *t*. *c*(*i*)*r*(*t*) = 0 refers the constant state and *c*(*i*)*r*(*t*) = 1 indicates the busy state, while [13]
(8)S={S1,…,SM}  with  M=2∑N i=1 R(i)

No states can be observed directly inside the POMDP infrastructure as in Figure 4; the collection of observations *Z*(*t*) ∈ *Z* is required to render the representation of the physical state. Then, observations are assumed with probabilistic behavior in which the observation function O is depicted as a probability distribution across the feasible observations *Z*(*t*), in action Φ(*t*), and final states *S*(*t*), which is represented by [27],
(9)O(S(t),Φ(t),Z(t))=Pr(Z(t)|Φ(t),S(t)),∀Z(t)∈Z,Φ(t)∈A,
where Φ(*t*) denotes the action set at *t*-th time slot. The attribute Φ(*t*) = {Φ(1)(*t*), Φ(2)(*t*), …, Φ(*N*)(*t*)} implies the action selected by POMDP development, and *S*(*t*) refers to the final state after implementing Φ(*t*). It is notable that Φ(*i*)(*t*) ∈ {0, 1} where 0 and 1 signify that carrier *i* is not applicable and accessed at time slot t. As the state transition and observation function are possible, it requires immediate reward *W*(*S*(*t*), Φ(*t*)) which can be accomplished by,
(10)W(S(t),Φ(t))=∑S^(t)∈SΓ(S(t),S^(t))∑Z(Z  O(S^(t),Φ(t),Z(t)) )C(Φ(t),S^(t)) 

### POMDP-Enabled Policy-Based Spectrum Allocation (PPBSA)

As a result of the implementation of the suggested POMDP in LTE-A small cells, the channel status indicator on the dispersed spectrum is evaluated considerably by sensing the frequency carriers under consideration [14,21]. To accomplish high system throughput, a contention approach has been allocated in the decision-making process mentioned in Algorithm 3, to access alternate systems-distributed channels. The two influential factors in selecting target-shared channels are channel state indicator (CSI), and number of contenders (NOCs). Here, the NOCs define the overall count of determined SUs and access to a similar spectrum. As a result, taking NOCs into account lowers SU collisions on the shared spectrum. The power and channel assignments are made to enable the small cell-enhanced node B (SeNB) to function as numerous UEs [28,29] without interference from small cells as in Figure 5.
**Algorithm 3.** Channel Selection and Optimal Power Allocation Policy1: **Input**: Channel State Identifier, Channel Occupancy Predictor, Belief State and Number of SUs
2: **Output**: Channel Selected, Updated CSI
3: Set Wmax = 0 
4: **for i** = **1 to N do**

5: Set Optimal carrier Φ(i)(t) = 1 and Φ(j)(*t*) = 0, ∀j ∈ {1, …, N}, j ≠ i, where Φ(*t*) = {Φ(1)(*t*), Φ(2)(*t*), …, Φ(N)(*t*)}
6: Derive Optimal Power allocation P(i) × (t) and A(i) × (t) 
7: Compute Expected Idle time of channel and probability that small cell reserves channel r) 
8: Compute reward and set Wmax = max{Wmax, W(i) (Φ(i)(t), P(i) × (t), A(i) × (t))}
9: **if** Wmax = W(i) (Φ(i)(t), P(i) × (t), A(i) × (t)) then
10: Set Optimal carrier Φ × (t) = Φ(*t*) 
11: **end if**
12: **end for**
13: Set update for Channel Selection A × (t) = {A(1) × (t), A(2) × (t), …, A(N) × (t)}

## 6. Results

### 6.1. Spectrum Analyzer Specification

The spectrum analyzer has been configured with the parameter setting as specified in Table 4 for the use cases.

### 6.2. USE CASE>>GSM [941.2 MHz–949.6 MHz]

The data are analyzed in Figure 6 and Figure 7 using several sample windows, each with 1024 discrete points. To reduce overlap between subsequent observation vectors, a block of 512 points was eliminated between each pair. The resolution bandwidth for each window is of value 10 MHz and is balanced on 942 MHZ, resulting in 9.7 kHz guard space between each pair of points captured every 10 μs. Parallel to our allocation method, we did an analytical investigation utilizing the open-source project “bit-gsm”, which allowed us to get an a priori understanding of the occupied PU considered territory.

With the Hanning window weighted by 4096 non-overlapping windows of 1024 received samples, the Welch technique was used to estimate the power spectral density at each acquisition point shown in the graph below. The GSM channel’s core frequency is shown by the vertical red lines. The PSD distribution implies tiny idle bands between 940.2 and 941.2 MHz, as well as a 4 kHz idle band for guard channels between 945.6 and 949.6 MHz.

When the number of SUs rises, the system’s total utility drops [30,31]; similarly, when the number of functioning PUs increases, the system’s average SU arrival declines. To minimize searching delays, it is ideal to do a limited number of channel searches before moving on to the next available channel. When the inter-sensing interval is minimal, the switching rate of the channels stays high [29,30]. This is because sensing occurs in a relatively small period of time, resulting in significant overhead and increased channel switching. As a result, detecting a signal with certainty is difficult, necessitating the addition of additional channels. Knowing which channels are most likely to be idle reduces channel switching and search times. Our channel selection algorithm picks channels for sensing intelligently. This significantly minimizes the number of channel shifts required, since only those channels with the greatest probability of being free are chosen for sensing. Channel switching diminishes as the inter-sensing interval increases, as seen in Figure 8. When sensing is carried out over an extended period of time, the results are more certain and likely to be accurate. What is critical in this case is to identify idle channels. If only idle channels are chosen, there is no need to move between several channels, resulting in fewer instances of channel switching [32,33]. The length of the sensing operation has a direct effect on network performance. A channel that has been sensed for a prolonged period of time slows data transmission, since data transmission may occur only once sensing is complete.

### 6.3. USE CASE>>VHF [180 MHz–270 MHz]:

Using a resolution of 30 kHz, the spectrum analyzer was able to identify fixed voltages as in Figure 9. Three 90 MHz segments with 3001 continuous channels were used to measure the whole spectrum from DC to 300 MHz. Typical VHF low-end broadcasts often have a bandwidth of 30 kHz, which is wider than 30 kHz. The result for the 180–270 MHz frequency range is shown in the graph below. There is a clear depiction of broadcast television channel 10 in this image (193.25 MHz, 196.83 MHz, and 197.75 MHz). There is a strong correlation between the mean and maximum PSDs, which indicates that the channels that are most often utilized have a high frequency. Following that, we make an effort to quantify spectral occupancy. Notably, huge sections of the spectrum are virtually always occupied by extremely powerful, very broadband transmissions—particularly, broadcast television and FM stations [34]. Clearly, incorporating such sections of the spectrum in computations of spectral occupancy is of little use.

It should be emphasized that the spectrum occupancy in the observations presented here is actually sparse. There is virtually little indication of sustained activity greater than 87 dBm per 30 kHz. For a rural environment, the frequency sample size taken is 25–90 MHz. PSD value of 100 dBm is a good sample. Frequency-agile operation in the metropolitan area may be accomplished by using 31 to 60 MHz and 141 to 180 MHz as feasible possibilities. These bands had 40 and 67 openings with bandwidths ranging from 29 kHz to more than 2.9 MHz. This band’s statistics may be affected by improvements in measurement sensitivity, which is expected. Furthermore, it should be remembered that broadcast television stations operating on many channels may be dominant in particular locations. Nonetheless, these are generally favorable results for the future of cognitive radio with frequency agility. The measurements provided are imperfect, and further effort will be required to provide a credible verdict. Next generation measurements should focus on long-term (weeks) observation at several sites with a time-frequency resolution of 1kHz in 1 msec. An evaluation of the feasibility and expected capacity of cognitive radio networks operating in the unallocated spectrum without interfering with major users must meet these severe conditions [35].

The following are the important stages in the measuring process:As though they were broadcasting alone, each linked SU transmits at the highest allowable power associated with the appropriate channel.If the television reception is compromised, reduce the transmit power of one or more secondary users until no more visual distortion is seen.Calculate the weighted aggregate power by recording the transmit power of each SU.

As anticipated, the weighted aggregate power is always equal to or less than the intended television signal level; hence, the suggested model is validated. The following graphic illustrates how, as the number of concurrently accessible channels rises, the maximum acceptable received interference in each adjoining channel decreases. One consequence of the accumulative impact of multichannel interference is a reduction in the amount of available TV white space for many secondary users [36]. The collective impact of neighboring channel interference is readily apparent, where a safe area is defined as the set of interference values that the SUs may create without interfering with TV reception. When the aggregate impact of adjoining channel interference is considered, we can plainly observe how the safe zone diminishes as in Figure 10.

Contrary to industrialized nations, a significant percentage of the TV band spectrum remains underutilized. Even with conservative criteria, the findings indicate that all 15 channels are free in at least 56.27% of the region. The average amount of accessible television white space was determined using two methods: first, from a protection and pollution standpoint, and second, from the perspective of FCC restrictions [36]. Both approaches indicated that the average accessible TV white space in the UHF television spectrum is more than 100 MHz as in Figure 11. A method for reassigning TV transmitter frequencies was devised in order to free up unneeded spectra. Eight television channels (about 64 MHz) were determined to be adequate to cover the current UHF/VHF television bands. In the future, we want to investigate appropriate laws for the TV white space to allow inexpensive broadband access.

It is pointed out that the overhead data rate is reduced by enhancing the sensing time. The reason for data rate limitation is due to the limited transmission time which is occupied by prolonged sensing duration. As a result, it demonstrates that the sensing time is described as 1 ms. In addition, it is observed, as, in Figure 12, that our projected PPBSA approach performs better than alternate models under sparse as well as dense scenarios.

### 6.4. Inferences

Two case studies have been analyzed for frequencies GSM 900 and the TV VHF band. Both the case studies have taught that the pattern of usage of the primary user, the guard space analysis, the pattern of the secondary user, and the occupancy criteria are considered to be the same. A POMDP model has been developed to understand the usability of spectral white space [34] by the secondary user and they are prioritized in a proposed swotting model database. The prioritized secondary users are assigned ranking through the rewards generated by the models. An optimal mechanism of spectrum transferred from the primary to the secondary user and the handoff from secondary to primary user need arises and is experimented with the statistical analysis as in Figure 13.

## 7. Conclusions

Various real-time bands were analyzed and the proposed elite channel allocation and mapping algorithms were tested to offer service to the secondary users requesting a single service. The elite-CAM is enhanced as policy and the policies are imposed on a policy engine. The policy-based spectrum allocation has been developed and implemented for ranking and rewarding the static environment. The policy engine is suggested for configuration with the spectrum management function of 5G CORE specification. The continuously changing behavior of the primary users and the users who do not fit in the ranking mechanism has not been experimented with. Therefore, it is not preferable that cognitive radio users once tested with a particular model prolong for a constant period of time. For a fast-changing dynamic environment, the spectrum allocation strategy should be fine-tuned. In addition, the handoff will be another major issue when the PU arrives back to the channel, and as future work, addressing dynamic spectrum allocation in 5G and beyond 5G networks is suggested.

## Figures and Tables

**Figure 1 sensors-22-05011-f001:**
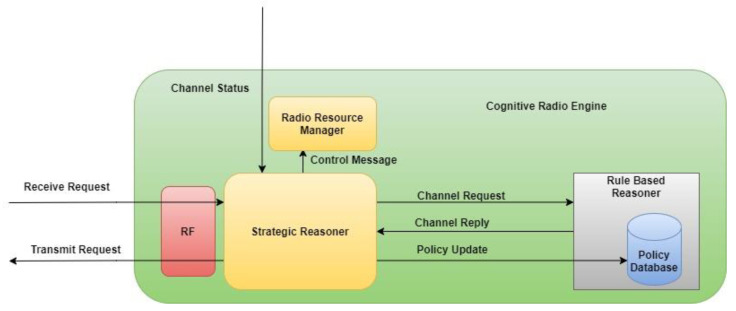
Policy Engine.

**Figure 2 sensors-22-05011-f002:**
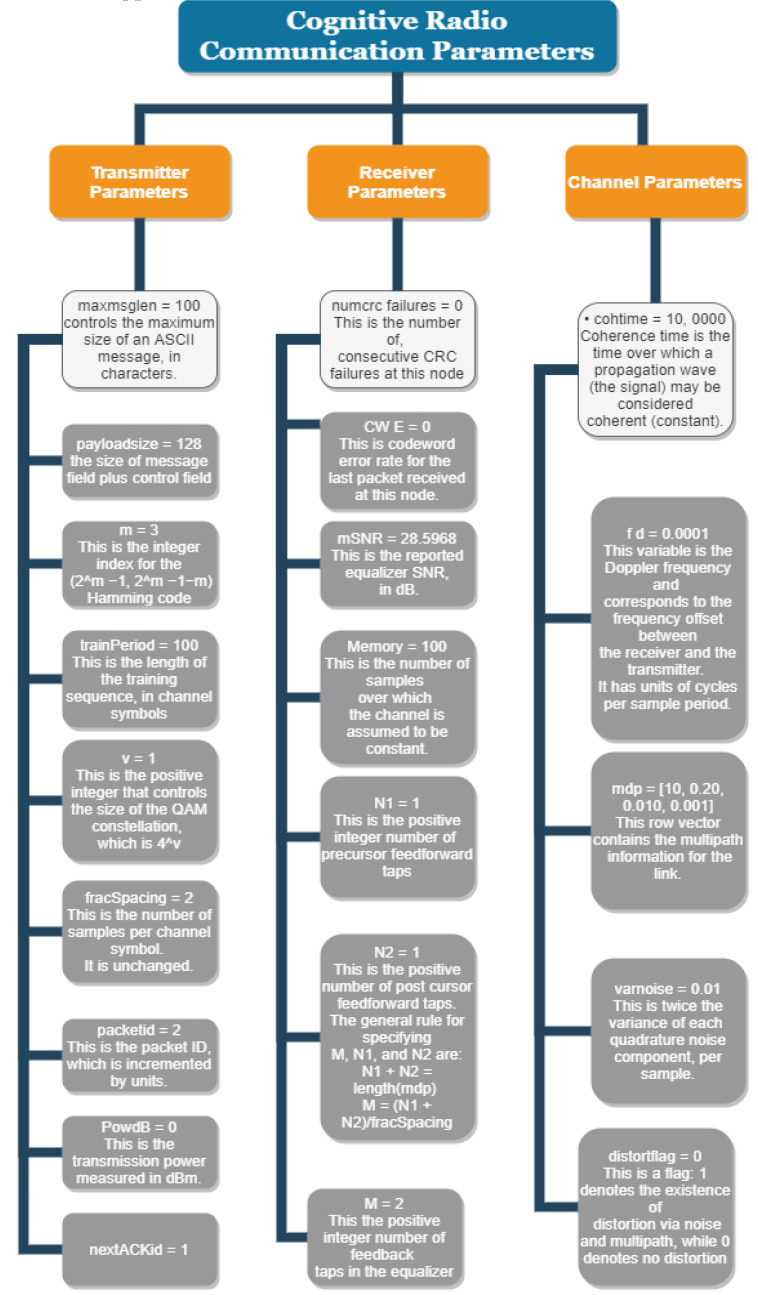
Cognitive Radio Parameters.

**Figure 3 sensors-22-05011-f003:**
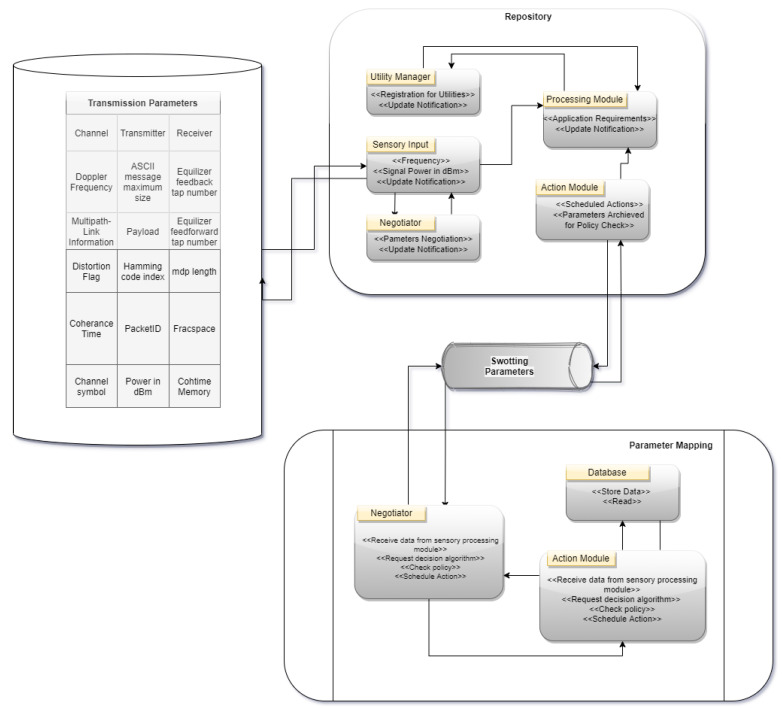
Swotting for cognitive radio parameters.

**Figure 4 sensors-22-05011-f004:**
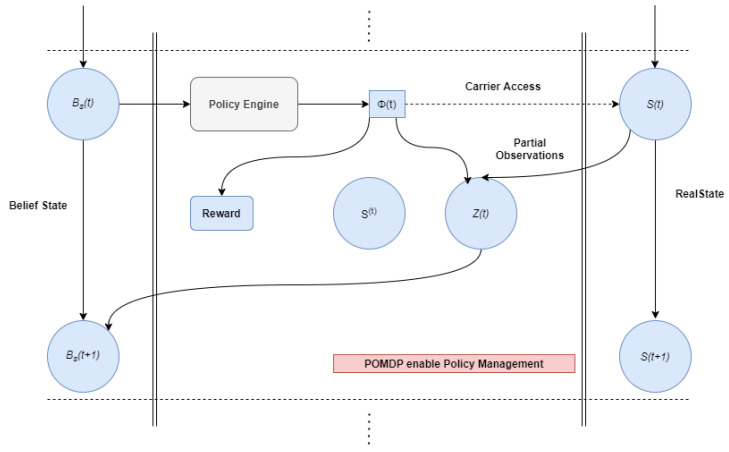
POMDP-enabled policy management.

**Figure 5 sensors-22-05011-f005:**
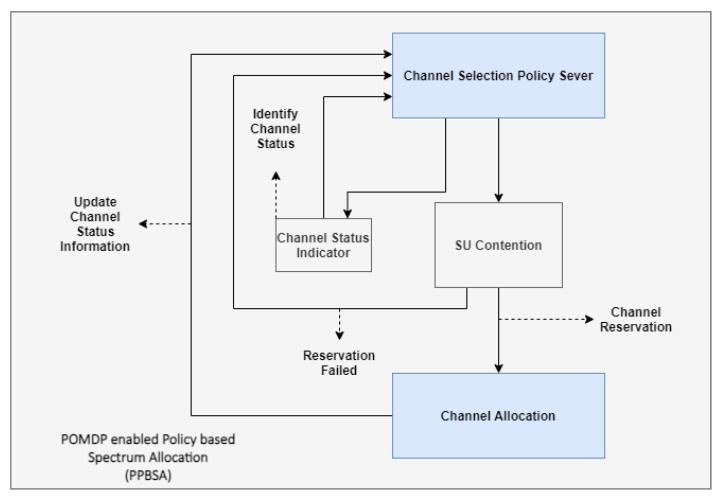
POMDP-enabled policy-based spectrum allocation (PPBSA).

**Figure 6 sensors-22-05011-f006:**
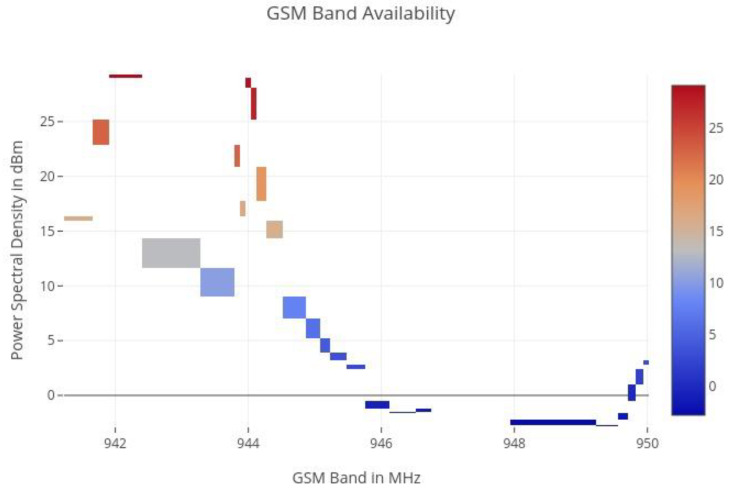
GSM 900 Band Availability.

**Figure 7 sensors-22-05011-f007:**
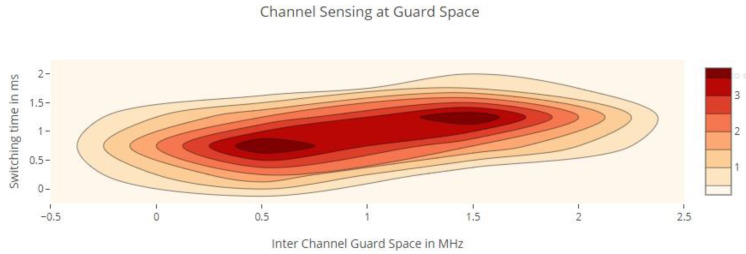
Channel switching time over guard space.

**Figure 8 sensors-22-05011-f008:**
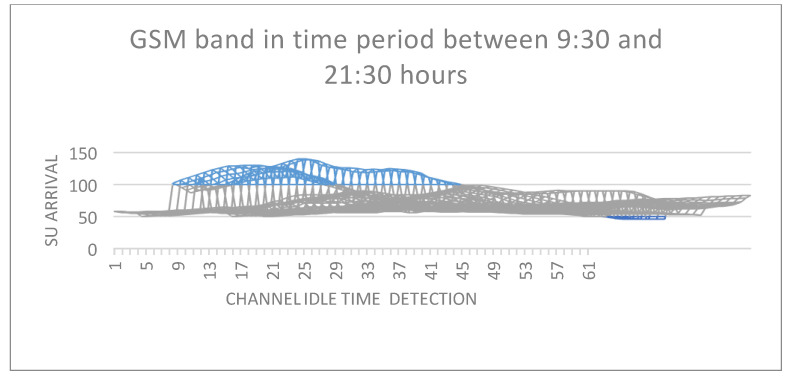
SU arrival over GSM 900.

**Figure 9 sensors-22-05011-f009:**
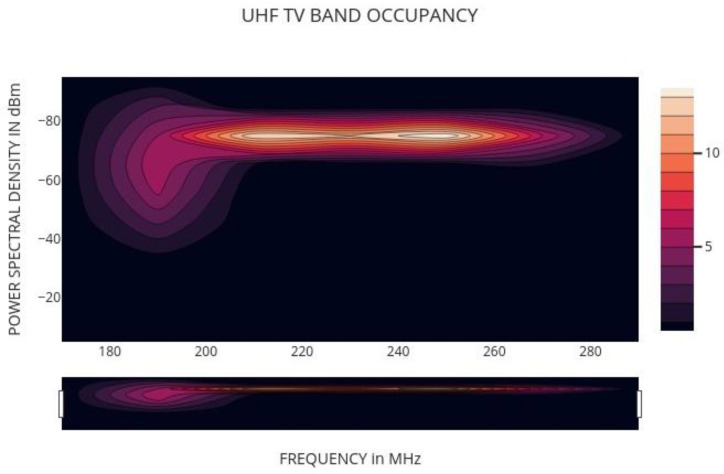
UHF band availability.

**Figure 10 sensors-22-05011-f010:**
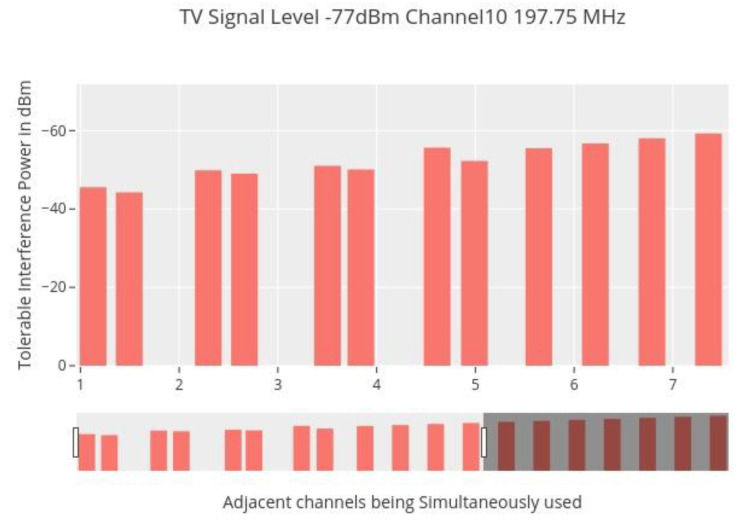
Adjacent channel interference.

**Figure 11 sensors-22-05011-f011:**
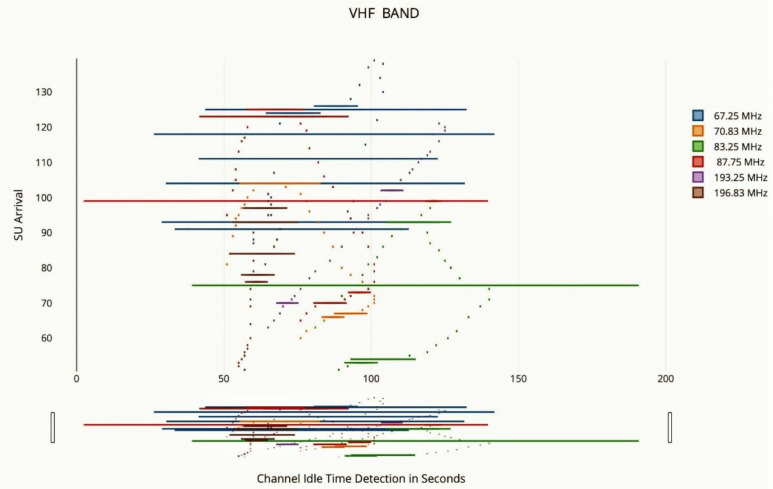
SU arrival over VHF band.

**Figure 12 sensors-22-05011-f012:**
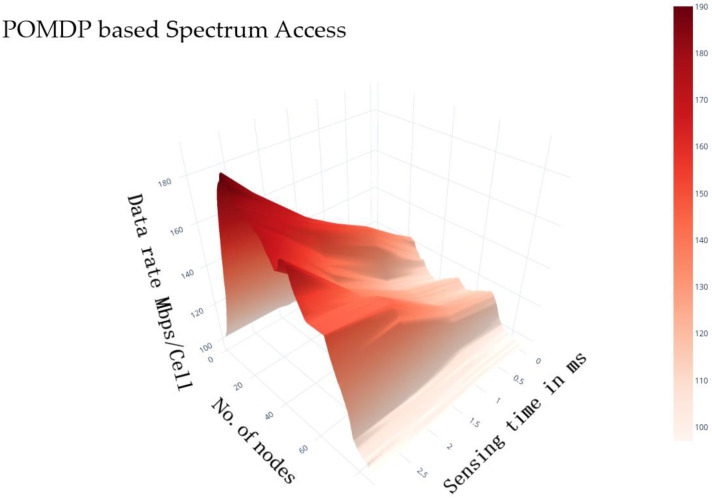
POMDP-based spectrum access by SU.

**Figure 13 sensors-22-05011-f013:**
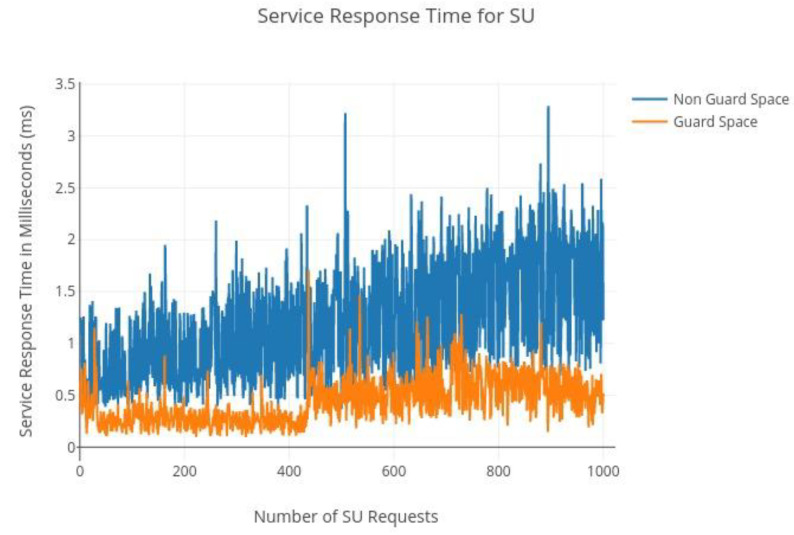
Service response time.

**Table 1 sensors-22-05011-t001:** Comparison Chart for Related Work.

Reference	Proposed Technique	Focus	Network	Simulation/Frequency Band	Parameters Improved
[1]	A detailed survey on spectrum sensing technique in 5G.	Sensing	Distributed	-	Spectrum trading and leasing Multi-user interference
[2]	A CR-enabled NOMA network capable of wireless data and power transmission at the same time.	Sensing	Distributed	10^6^ iterations Monte Carlo simulations	Power outage probability
[3]	One of SenPUI’s key issues, detecting and Primary User Interference, is addressed by a unique cognitive radio algorithm.	Sensing	Centralized	Real-time implementation with IEEE 802.15.4	Throughput Primary User Interference
[4]	Models for analyzing and evaluating sharing procedures under a wide variety of situations were provided.	Sharing	Distributed	MATLAB	PU activity SU Arrival
[5]	In cognitive radio networks, the hidden Markov model (HMM) was used for opportunistic spectrum access (OSA) through cooperative spectrum sharing.	Sharing	Distributed	HMM in OSA	Detection Probability
[6]	The CR networks may now be made aware of the needs of unlicensed users through a new method that reduces sensing latency.	Sensing	Distributed	Q-learning for different radio access techniques	Sensing Latency
[7]	Two machine learning (ML) approaches that have been developed to increase spectrum sensing performance are k-nearest neighbors and random forest.	Policy	Distributed	Energy detection using k-NN and RF Algorithms	Energy Detection
[8]	Massive MIMO cognitive radio underlay user selection was proposed using a QoS-aware technique.	Policy	Distributed	Deployed a DNN with MIMO CR	Loss Function Success Rate Average transition time

**Table 2 sensors-22-05011-t002:** Cognitive Radio Parameters.

PARAMETERS		
Channel	Transmitter	Receiver
Doppler Frequency f^d^ (0.0001)	ASCII message maximum size Maxmsglen (100)	Equalizer feedback tap number N1
Multipath-Link Information md_p_ (10, 0.2, 0.001, 0.010)	Message + control message Payload (128 bits)	Equalizer feedforward tap number N2
Variance of Quadrature noise varnoise (0.01)	Hamming code index M (3)	N1 + N2 = md_p_ length
Distortion Flag distort flag (0 for no distortion and 1 for noise)	Samples/Channel symbol Fracspace (2)	Fracspace (2)
Coherence Time Cohtime (100,000)	PacketID	Cohtime Memory
	ACKID	
	Power in dBm maxpower	

**Table 3 sensors-22-05011-t003:** Cognitive Channel Allocation Policies.

Cognitive Channel Allocation Policies				
Scenario	Pu is occupying the spectrum. SUs are in queue.	The spectrum is free, PU is idle and not utilizing the spectrum. SU is in need.	PU has left the spectrum free, No. of the competing SUs is more for the same spectrum.	PU has been left with a fading channel. Spectral density is low. The secondary user arrival rate is Poisson.
Policy	Wait	Allocate	Assign by Rank	Random wait
Reason	As the history of the PU activity and the requirement of the SU is known, mapping is already done, the mapping table is verified, and then policy 1 is triggered.	The spectrum is freed by PU. SU is in need of the spectrum. Spectral density is good. SU’s requirement lies within the availability of the spectrum. SU’s Qos is also satisfied by the network and spectrum parameters.	SUs are prioritized for their effective utilization of the spectrum and their active participation without wasting the spectrum. Their spectral density and utilization factor are the criteria for the decision.	The spectrum analyzer has to detect the quality of the channel by sending a few random packets at different time intervals.

**Table 4 sensors-22-05011-t004:** Spectrum analyzer specification.

Parameters	Range Setup 1	Range Setup 2	Range Setup 3
Spectrum Frequency	99 kHz–3 GHz	100 MHz–1000 MHz (UHF) GSM 900 MHz 2.4–2.5 GHz (ISM band)	50 MHz–4400 MHz [GSM 900, GSM 1800]
Duration	12 h [6 a.m.–6 p.m.]	24 h	24 h
Instance	60	90	120
Sweep Time [Frequency Span = 0 Hz]	1 millisecond–100 s	Auto	Auto
Sweep Time [Frequency Span > 0 Hz]	20 milliseconds–1000 s	Auto	Auto
Bandwidth for Video	10 Hz–1 MHz	100 kHz	10 MHz
Interface	RS232	RS232	RS232
Bandwidth Resolution	10 Hz–1 MHz	100 kHz	10 MHz

## Data Availability

The data used to support the findings of this study are available from the author (ihra@kunsan.ac.kr) upon request.

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
