# Peer review of "Elite-CAM: An Elite Channel Allocation and Mapping for Policy Engine over Cognitive Radio Technology in 5G"

_sensors, 2022, doi:10.3390/s22135011_

Round 1
Reviewer 1 Report
1. Figures 2 and 3 are clear to see.
2. It is not easy to know what are the parameters mean in Table 1 and the relationship with the key idea of this paper.
3. For experiments, where are data come from? This is the key of this paper. If the data is not sufficient and effective, it is not easy to justify whether the proposed method is effective.
4. Figure 11 seems not the experimental results and should be given sufficient citation.
Author Response
Report
Reviewer 1
- It is not easy to know what are the parameters mean in Table 1 and the relationship with the key idea of this paper.
Response: The parameters presented in Table 1 represent the exhaustive set of parameters for a Cognitive Radio enabled environment. The algorithms for Channel Selection and Allocation, and the Parameter Mapping retrieve the parameters from the exhaustive set presented in Table 1.
- For experiments, where are data come from? This is the key of this paper. If the data is not sufficient and effective, it is not easy to justify whether the proposed method is effective.
Response: The data specification for the spectrum analyzer is given in Use cases 6.1 and 6.2. With the identified 12-hour-spectrum occupancy data, the PoMDP enabled Policy Based Spectrum Allocation Engine is simulated with 10 carrier frequency bands, 5 reserved channel for hand-over, in LTE small cell environment using Matlab. With the exponential increase of secondary users at 5 instances in same network and dimensions
- Figure 11 seems not the experimental results and should be given sufficient citation.
Response: Figure 11 depicts the derived idle time with the observed spectrum occupancy data. It is a original chart with the formal observation of data
Reviewer 2 Report
The paper presents an interesting solution in dynamic spectru usage, and propose concrete Policy Engine with several case studies. The work certainly has a merit and contribution but the presentation, organization and technical level unable reader to properly grasp the underlying assumptions, principal ideas, details of proposed solution and thus the contribution of the paper.
In order to be therily and properly reviewed authors should reorganize and mostly rewrite the paper in order to clearly define motivation, contribution, realted work (that should be on Policy engine as the proposed Elite is and not to core cognitive radio network spectrum sensing) and state-of-the-art, more clearly write a description of the proposed solution (figures are unreadable, given algorithms are not referenced in text and are misplaced in relation to presentation flow, formatting, lanhuage, etc.). Finally, from current presentation it seems that Results section represent the input for the design of here proposed solution, and its placement after the description of the proposed Elite make a confusion.
Some suggestions in this process:
(1) The Introduction section should more clearly define motivation but also the main proposition and contribution of the paper, as well as the organization of the following text. Notice: The last part of the second paragraph in Introduction section is realtively uncelar - the tabulated form should be used if the text is conceived as a list of statements.
(2) The related work section should be rewritten and reorganized. This section should be focused to address the issues that are resolved by proposed solution, which is not clear in current version. The current version mostly deals with CRN spectrum sensing and th erealtion to devised Policy engine is not established.
- In current version all work is given in one paragraph with some part of these description not properly sequenced, i.e. " Assumption 2: In this situation, t ..." what is Assumption 1? to which work Assumption 2 relates? and so on.
- Is the last part dedicated to connect related work with proposed solution? It is not clearly stated, and the last part " Our neural network was able to rapidly learn the best user selection strategy ... " suggest that this last part defines a contribution of manuscript (which I am not sure to be correct). Also, formatting, punctation and language errors should be corrected.
(3) The section 3 more closely refers to related work for proposed solution / but it lacks references and overview, and just state the issues and broad form of Policy engine,
(4) The sections in which the proposed solution is described must be therilly rewritten, with proper definition and referencing to pictures, algorithms and so on. All relevant references used to define mathematical model should be stated.
(5) From thu surrent Result section it is not clear how these results support the claims on performance and quality of the proposed solution - are these inputs or outputs for proposed Elite engine?
Author Response
Reviewer 2
The Introduction section should more clearly define motivation but also the main proposition and contribution of the paper, as well as the organization of the following text. Notice: The last part of the second paragraph in Introduction section is relatively uncelar - the tabulated form should be used if the text is conceived as a list of statements.
Response: The introduction section has been appended with the details as suggested
The related work section should be rewritten and reorganized. This section should be focused to address the issues that are resolved by proposed solution, which is not clear in current version. The current version mostly deals with CRN spectrum sensing and the relation to devised Policy engine is not established.
- In current version all work is given in one paragraph with some part of these description not properly sequenced, i.e. " Assumption 2: In this situation, t ..." what is Assumption 1? to which work Assumption 2 relates? and so on.
- Is the last part dedicated to connect related work with proposed solution? It is not clearly stated, and the last part " Our neural network was able to rapidly learn the best user selection strategy ... " suggest that this last part defines a contribution of manuscript (which I am not sure to be correct). Also, formatting, punctation and language errors should be corrected.
Response: Thanks for your suggestion, we have processed and refined the content of the related work with a comparison chart and the related work has been appended with the works which justify the efficiency of policy management. The typograph errors has been corrected as suggested.
The section 3 more closely refers to related work for proposed solution / but it lacks references and overview, and just state the issues and broad form of Policy engine,
Response: The significance of policy engine has been stated in the section 3. The references have been cited as suggested
The sections in which the proposed solution is described must be theoretically rewritten, with proper definition and referencing to pictures, algorithms and so on. All relevant references used to define mathematical model should be stated.
Response: All the pictures used in the manuscript are original figures from the authors. The mathematical models have been cited as suggested.
From the current Result section it is not clear how these results support the claims on performance and quality of the proposed solution - are these inputs or outputs for proposed Elite engine?
Response: The proposed Elite engine is the framework of the contributed algorithms. The output of the PoMDP-enabled channel state and reward mechanism is the last in the sequence of the contributed algorithms. The output of service response time is considered as the outcome of the contribution.
Reviewer 3 Report
The contribution of the manuscript should be stated clearly at the end of the introduction and the advantages over related works should be pointed out.
The authors should enrich the introduction with state of the art references. In addition, the advantages of cognitive radio and spectrum utilization should be discussed and some references might be added:
[Ref 1] J. Mitola and G. Maguire, "Cognitive radio: making software radios more personal", IEEE Personal Communications, vol. 6, no. 4, pp. 13-18, August 1999.
[Ref 2] S. Haykin, "Cognitive radio: brain-empowered wireless communications", IEEE Journal on Selected Areas in Communications, vol. 23, no. 2, pp. 201-220, February 2005.
[Ref 3] A. Jamoos and A. Abdou, "Spectrum Measurements and Analysis for Cognitive Radio Applications in Palestine," 2019 6th International Conference on Electrical and Electronics Engineering (ICEEE), 2019, pp. 180-185, doi: 10.1109/ICEEE2019.2019.00042.
Many acronyms all over the manuscript such as (DAI, IoT, CR, SenPUI, PUI, ML, MIMO, SBS) are not defined.
The notations in equation (1) and (2) are not well defined. Check the correctness of these equations.
Figures 2, 3, 11, 12 are unclear.
The power spectrum density values in figure 6 are not realistic!.
Author Response
Reviewer 3
Date of this review
The contribution of the manuscript should be stated clearly at the end of the introduction and the advantages over related works should be pointed out.
The authors should enrich the introduction with state of the art references. In addition, the advantages of cognitive radio and spectrum utilization should be discussed and some references might be added:
[Ref 1] J. Mitola and G. Maguire, "Cognitive radio: making software radios more personal", IEEE Personal Communications, vol. 6, no. 4, pp. 13-18, August 1999.
[Ref 2] S. Haykin, "Cognitive radio: brain-empowered wireless communications", IEEE Journal on Selected Areas in Communications, vol. 23, no. 2, pp. 201-220, February 2005.
[Ref 3] A. Jamoos and A. Abdou, "Spectrum Measurements and Analysis for Cognitive Radio Applications in Palestine," 2019 6th International Conference on Electrical and Electronics Engineering (ICEEE), 2019, pp. 180-185, doi: 10.1109/ICEEE2019.2019.00042.
Response: Thanks for the suggestion. The journals has been added in the references as suggested
Many acronyms all over the manuscript such as (DAI, IoT, CR, SenPUI, PUI, ML, MIMO, SBS) are not defined.
Response: Thanks for your suggestion, we have processed and refined the acronyms as suggested
The notations in equation (1) and (2) are not well defined. Check the correctness of these equations.
Response: The notions are corrected as suggested
Figures 2, 3, 11, 12 are unclear.
The power spectrum density values in figure 6 are not realistic!.
Response: All the figures used in the manuscript are original figures from the authors. The power spectrum density values has been observed and incorporated in the contribution
Round 2
Reviewer 1 Report
accept
Author Response
Thank you for accepting our paper
Reviewer 2 Report
The authors have taken into account most of my previous comments and suggestions.
However, there are several minor issues that should be resolved:
- In Section 2. Related work, the whole section is given in the single paragraph. Some division in several paragraphs related to specific topics could improve clarity of th epresentation. Also, newly added Table1 should be mentioned in the text.
- In lines 221-226 the numering should be corrected.
- In Sections 3 and 4 all Tables, Algorithms and Figures should be mentioned in text and described. The quality of several Figures (e.g. 2, 3, 11, 12) should be improved.
- The results/information (as well as the description of important parameters for setting used to derive results) on Figures in sections 5 and 6 should be described in more details.
- Some references are not mentioned in the text, which should be corrected.
Author Response
The authors have taken into account most of my previous comments and suggestions.
However, there are several minor issues that should be resolved:
- In Section 2. Related work, the whole section is given in the single paragraph. Some division in several paragraphs related to specific topics could improve clarity of the representation. Also, newly added Table1 should be mentioned in the text.
Response: The related work section is segmented and correction has been highlighted as suggested
- In lines 221-226 the numbering should be corrected.
Response: Thanks for your suggestion, we have corrected the numbering
- In Sections 3 and 4 all Tables, Algorithms and Figures should be mentioned in text and described. The quality of several Figures (e.g. 2, 3, 11, 12) should be improved.
Response: All the images mentioned are redrawn by the authors. The algorithms and figures have been cited as suggested.
- The results/information (as well as the description of important parameters for setting used to derive results) on Figures in sections 5 and 6 should be described in more details.
Response: The spectrum analyser parameter setting is added as a table. The parameters have driven the observation and inferences in sections 5 and 6
Reviewer 3 Report
The authors have taken into account some of my previous comments and suggestions. However, the quality of Figures 2, 3, 11, 12 are still poor, they should be improved. In addition, the power spectrum density values in figure 6 are not realistic!. typical experimental values are from -60dBm to -100dBm. Furthermore, some of the listed references are not mentioned in the text.
Author Response
The authors have taken into account some of my previous comments and suggestions. However, the quality of Figures 2, 3, 11, 12 are still poor, they should be improved.
Response: Thanks for the suggestion. The mentioned figures are redrawn as suggested.
In addition, the power spectrum density values in figure 6 are not realistic!. typical experimental values are from -60dBm to -100dBm.
Response: Thanks for your suggestion. The PSD values specify the distribution and it has been highlighted in the table in section 6.
Furthermore, some of the listed references are not mentioned in the text.
Response: All the references are cited as suggested